# Effect of Harvest Date on Kernel Quality and Antioxidant Activity in *su1* Sweet Corn Genotypes

Tatjana Ledenčan [1], Daniela Horvat [1], Sanja Špoljarić Marković [2], Zlatko Svečnjak [3], Antun Jambrović [1,4] and Domagoj Šimić [1,4,*]

[1] Agricultural Institute Osijek, Juzno Predgradje 17, 31000 Osijek, Croatia; tatjana.ledencan@poljinos.hr (T.L.); daniela.horvat@poljinos.hr (D.H.); antun.jambrovic@poljinos.hr (A.J.)
[2] Croatian Agency for Agriculture and Food, Vinkovacka cesta 63c, 31000 Osijek, Croatia; sanja.spoljaric@hapih.hr
[3] Faculty of Agriculture, University of Zagreb, Svetosimunska cesta 25,10000 Zagreb, Croatia; svecnjak@agr.hr
[4] Centre of Excellence for Biodiversity and Molecular Plant Breeding, Svetosimunska 25, 10000 Zagreb, Croatia
* Correspondence: domagoj.simic@poljinos.hr

**Abstract:** In order to achieve a high-quality product, sweet corn should be harvested at the optimal stage of maturity. The taste of sweet corn depends largely on the kernel moisture (KM) and total sugars (TS) content while its health properties are attributed to the content of total phenols (TPH) and antioxidant activity (AO). This study aimed to estimate quality in sugary (*su1*) sweet corn hybrids based on the maximum content of kernel chemical compounds and the content change during maturation. During two growing seasons, five harvests by year were made at 2-day intervals from 17 to 25 days after pollination (DAP). KM decreased among hybrids from 6.6% to 14% over an eight-day period, or 0.8% to 1.8% per day. TS content was the highest at 17 DAP (16.5–28.7 mg/g DW) and decreased during maturation from 27% to 67%. Hybrids differed significantly in TPH at 17 DAP (204.4–357.1 mg GAE/100 g DW). TPH change during maturation ranged from the no significant differences up to a 29.3% decrease. DPPH- radical scavenging activity ranged among genotypes from 52.5% to 74.9% inhibition at 17 DAP and significantly increased during maturation. A well-defined hybrid-specific harvest window is necessary to maintain kernel quality in *su1* sweet corn.

**Keywords:** sweet corn; maturation; kernel moisture; sugars; phytoglycogen; total phenols; DPPH

## 1. Introduction

Sweet corn is a widespread and economically important vegetable that is used for fresh market or processing either frozen or canned. Sweet corn is the result of recessive mutation of field (flint/dent) corn in the genes which control conversion of endosperm sugar to starch. Among the many mutants known to alter the composition of endosperm sugars, sugary (*su1*), shrunken (*sh2*) and sugary enhancer (*se1*) are the most commercially represented [1]. Homozygous *su1* reduces starch levels and increases the level of sugars and water-soluble polysaccharides (phytoglycogen) that give the endosperm a smooth creamy texture characteristic of "normal" or "standard" sweet corn [2,3].

At optimum sweet corn harvest maturity, kernels are sweet, plump, milky, and tender [4]. Kernel chemical composition changes quickly during maturation, and therefore harvesting sweet corn at the proper stage of maturity is essential to ensure a high eating quality. Sweet corn eating quality is determined largely by kernel moisture at harvest and total sugars content. Kernel moisture is a basic parameter for determining the technological maturity of sweet corn [5]. For sugary genotypes, optimal kernel moisture at harvest ranges from about 76% towards 70%, depending on the fresh or

processing use [4,6]. Sugary genotypes are characterized by very fast kernel moisture loss and consequently narrowed harvest window [6].

Endosperm sugars determine flavor, the primary component of fresh quality associated with consumer preference [7–9]. Sensory evaluations show a high correlation between the sweetness of sweet corn and the sugar content in the endosperm, especially sucrose [8,10–12]. Usually, genotypes reach maximum endosperm sugar content 18 to 20 days after pollination and further maturation reduces the sugars content, which results in a decrease in kernel sweetness [6]. Due to fast conversion of sugars to phytoglycogen and starch, sugary genotypes retain the achieved amount of sugar for a much shorter time during maturation than other sweet corn mutants [8,13–15].

Recently, sugary sweet corn hybrids have constituted a smaller share in production compared to *sh2* and *se1* hybrids, mainly due to significantly lower sugar content and rapid loss of quality during maturation and after harvest [16]. However, in sweet corn produced for fresh consumption, production in earliness and late season is highly important for providing the market with fresh crops. Extended growing seasons cause sweet corn to be more exposed to stress. Sugary hybrids have better seed emergence and early vigor in cooler planting conditions, as well as better tolerance for temperature and water stress during pollination and maturation, which is why they are still widely used by smaller farmers for local fresh markets production [6].

Beside a taste of sensory enjoyment, sweet corn provides high nutritional value and is becoming an increasingly important component of a healthy diet. Sweet corn health benefits are mainly attributed to the high abundance of non-nutrient phytochemicals, especially phenolic, and their potent antioxidant activity. Bioactive properties of phenolic compounds are associated with a reduced risk of many chronic disorders such as diabetes, cancers, and cardiovascular diseases [17–19]. A study by Song et al. [20] in commonly consumed vegetables, showed no significant differences in total phenols content among sweet corn, carrot, potato, white onion, green pea, and tomato. Based on total phenols content and amount of daily consumption, sweet corn was estimated to be the third highest total phenols provider (after potatoes and tomatoes) among the daily consumed vegetables in the United States [21]. Antioxidant activity in fresh sweet corn (49.3% inhibition of 1,1-diphenyl-1-picrylhydrazyl (DPPH) radical), was found to be higher than in spinach and peas [22].

The objective of this study was to examine the effect of maturation on the chemical components of kernel quality and antioxidant activity in *su1* sweet corn hybrids. It is important within well-defined optimum harvest window for achieving best eating quality, to identify the peak at which *su1* sweet corn has the maximum health-benefiting properties.

## 2. Materials and Methods

### 2.1. Plant Materials and Field Experiments

Field experiments were conducted at the experimental station of Agricultural Institute Osijek, Croatia (45°33′ N, 18°41′ E) in two years (2018 and 2019) in a three-year crop rotation (wheat-soybean-maize) where soybean was a previous crop. Location is characterized by semiarid climate with 30-year (1981–2010) averages for daily air temperature of 10.8 °C and total precipitation of 610 mm. Usual local crop management practice for high-yielding sweet corn was applied according to the local rain-fed regime. Five commercial sweet corn hybrids (OS 230su, OS 247su, OS 254su, OS 255su and OS 256su) developed at Agricultural Institute Osijek were grown in four-row 6 m long plots, containing 25 plants per row (65,000 plants/ha). The experimental design was randomized complete block design (RCBD) in two replications. Planting dates were 27 April in 2018 and 26 April in 2019. To avoid xenia effect and to ensure uniform maturity and genetic purity, plants in the two central rows were self-pollinated by hand. Pollination took place from 29 June to 2 July in 2018 and from 1 July to 3 July in 2019 depending on particular

genotype. At a 2-day interval from 17 to 25 days after pollination (DAP), five successive hand harvests were made. As a sample, five randomly chosen self-pollinated ears in each replication were harvested. All samples were taken early in the morning and transported directly to the processing laboratory.

Average monthly air temperatures and monthly precipitation during the field experimentation are presented in Table 1. In 2018, the sweet corn growing season April and May was hotter and drier compared to 2019 and the long-term average. A higher total amount of precipitation in June was recorded in both investigated years. Large differences in the amount of precipitation in July between the two growing seasons had an impact on sweet corn maturation.

**Table 1.** Monthly climate data related to the evaluated sweet corn growing seasons and a long-term average from 1991 to 2020 (LTA) in Osijek, Croatia.

| Month | Precipitation (mm) | | | Temperature (°C) | | |
|---|---|---|---|---|---|---|
| | **2018** | **2019** | **LTA** | **2018** | **2019** | **LTA** |
| April | 21 | 69 | 51 | 17.0 | 12.8 | 12.3 |
| May | 27 | 151 | 74 | 20.6 | 14.0 | 17.0 |
| June | 127 | 131 | 82 | 21.7 | 23.1 | 20.6 |
| July | 132 | 57 | 64 | 22.5 | 22.6 | 22.3 |
| Total precipitation | 307 | 408 | 271 | | | |
| Average temperature | | | | 20.4 | 18.3 | 18.1 |

Source: Croatian Meteorological and Hydrological Service [23].

### 2.2. Sample Preparation for Biochemical Analysis

Field samples were placed in a refrigerator at 4 °C and within half an hour the ears were peeled, part of the tip and butt of each ear was discarded, and kernels were hand-cut from the cob. Extractions of sugar and phytoglycogen were performed from fresh kernels, while samples for the extraction of total phenols were lyophilized (ALPHA 1-2 LD, Christ, Osterode am Harz, Germany).

### 2.3. Determination of Kernel Moisture at Harvest

The 5 g of fresh sweet corn kernels was thoroughly homogenized and dried at 105 °C (Memmert GmbH + Co. KG, Schwabach, Germany) to a constant weight, reweighing and the calculation of the percentage of kernel moisture was made.

### 2.4. Determination of Sugars and Phytoglycogen

An amount of 5 g of a previously well mixed and homogenized fresh sample was loaded in a micro-tube for extraction of sugar and phytoglycogen by different extraction solvents according to some modification method of Simla et al. [24]. Sugars were extracted with 90% ethanol at 80 °C (Shaking water baths GFL 1092, Germany). One milliliter of 90% ethanol was added in the micro-tubes, well shaken (Vibromix 204 EV, Tehtnica, Železniki, Slovenia) and centrifuged at 15,000 rpm for 7 min (Universal 320R; Hettich, Tuttlingen, Germany). The extraction process was repeated three times and collected supernatants were stored at −20 °C until further analysis.

Soluble sugars composition was analyzed by isocratic HPLC system series 200 equipped with refractive index detector and TotalChrom Navigator software (Perkin-Elmer, Waltham, MA, USA). Prior analysis, the extract was filtered through 0.45 μm nylon membrane filter. 20 μL of extract was injected and sugars were separated on a MetaCharb Ca Plus column (300 × 7.8 mm) at 90 °C. Degassed water with a flow rate of 0.5 mL/min was used as mobile phase. To generate calibration curves, three concentrations of glucose, fructose, sucrose and galactose (5, 10 and 15 mg/mL) were subjected to HPLC analysis. Sugars were identified by comparison of retention time with to those of standards and quantified by internal standard method using the same amount of galac-

tose in each sample. The content of glucose, fructose and sucrose were expressed as mg/g of dry weight (DW). The total sugar content was estimated by summing the amount of each sugar.

For phytoglycogen extraction, the remaining pellet after extracting sugars was used. On pellet, one ml of 10% ethanol was added in micro-tube in water bath at 80 °C, well shaken and incubated in the refrigerator at 4–5 °C for 12 h. After that, the samples were centrifuged 10 min at 15,000 rpm. The extraction process was repeated three times for better release of phytoglycogens and collected supernatants were stored at −20 °C until further analysis. Quantification of phytoglycogen was carried out at 490 nm (Specord 200, Analytic Jenna, Jenna, Germany) using phenol-sulfuric colorimetric method [25] with D-glucose as a standard.

### 2.5. Determination of Total Phenols

The total phenols from sweet corn kernels were determined with Folin-Ciocalteu reagent according to Singleton and Rossi [26] with some modifications. A quantity of 1 g of lyophilized sample was mixed with 10 mL of 1% HCl in pure methanol (purity grade 99.8%). The mixture was sonicated (Sonorex Digitec, Bandelin, Berlin, Germany) for 1 h and centrifuged (Universal 320R; Hettich, Germany) at 4000 rpm for 10 min. Collected supernatant was used for determination of total phenols content and their antioxidant activity. All samples were extracted in duplicate.

For total phenols content, 0.1 mL of extract was reacted with 5 mL of Folin-Ciocalteu reagent (1:10) and 0.9 mL of dH$_2$O followed by adding 4 mL of 7.5% Na$_2$CO$_3$ solution. The thoroughly shaken mixture was incubated for 2 h in dark at room temperature. Total phenols were quantitated by spectrophotometric method at 765 nm (Specord 200, Analytic Jenna, Jenna, Germany). All measurements were performed in triplicate. The external calibration curve of gallic acid (0.05–1 mg/mL) was used for total phenols quantification and results were expressed as gallic acid equivalent (GAE) per 100 g of dry matter.

### 2.6. Antioxidant activity assay

Antioxidant activity was determined using free radical 2,2-diphenyl-1-picrylhydrazyl (DPPH) radical scavenging assay. Antioxidant activity was measured using a modified version of the method of Brand-Williams et al. [27]. For each measurement, 0.2 mL was taken from the sample extract and a mixture was formed by adding 1 mL of a 0.5 mmol/l methanol solution of DPPH and 2 mL of methanol. After incubation for 30 min in a dark place, the absorbance was determined at 517 nm. Like the sample, the blank was also incubated for 30 min in the dark. The procedure was repeated three times. The percentage of inhibition of free radical DPPH in percent (%) was calculated against blank:

$$\% \text{ Inhibition} = (1 - (A \text{ sample}_{t=30}/A \text{ blank}_{t=0})) \times 100$$

### 2.7. Statistical Analysis

Field trials conducted over two growing seasons were arranged in a strip-plot design with two replicates. The genotypes were laid out in vertical strips in a randomized complete block design, whereas harvest dates were laid out in strips horizontally in the same replication. For statistical analysis, analysis of variance (ANOVA) was applied using the proc mixed procedure in SAS 9.4 (SAS Institute Inc., Cary, NC, USA) [28]. The main effects of harvest day, genotype, and growing season were treated as fixed, and replications as a random factor. In ANOVA, F-test was applied, while for mean separation Fisher's protected LSD test was used at the 0.05 and 0.01 probability level.

## 3. Results and Discussion

### 3.1. Kernel Moisture Change during Maturation

Kernels moisture content strongly affects kernels texture and tenderness, and for the sweet corn processing industry, kernel moisture represents an indicator of technological maturity [29]. The optimum kernel moisture for harvesting depends on whether sweet corn intended use is for fresh market, freezing or canning and generally it falls within interval from 70% and 76% [4,6]. Based on analyses of variance (Table S1), a significant harvest date, genotype and year effect on the kernel moisture at harvest was found. Kernels moisture change during sweet corn maturation is shown in Table 2. Moisture content significantly decreased with maturation, dropping among genotypes from 6.6% (OS 255su) to 14% (OS 230su) over eight-day period or 0.8% to 1.8% per day. All hybrids had higher grain moisture loss in the 2019 growing season, which was influenced by the absence of precipitation in July through all harvest dates in July [30,31]. Considering 70% kernel moisture as the lowest optimal moisture content for harvesting, the length of optimal harvesting period varied among hybrids up to two harvest dates. The narrowed harvest window was obtained for hybrid OS 230su in both growing seasons. Hybrid OS 255su showed as a letter maturation group with the highest kernel moisture at all harvest dates, and the data suggest that its optimal harvesting period could be extended beyond 25 DAP.

**Table 2.** Kernel moisture at harvest (%) in five sugary (*su1*) sweet corn hybrids at five harvest dates 17–25 days after pollination (DAP) in two growing seasons (Osijek, 2018 and 2019). The different lowercases show significant differences among the harvest dates within hybrids ($p < 0.05$). The different capital letters show significant differences among the hybrids ($p < 0.05$).

| Year | DAP | OS 254su | OS 255su | OS 256su | OS 247su | OS 230su |
|---|---|---|---|---|---|---|
| 2018 | 17 | 78.3 ± 0.1 [aAB] | 79.2 ± 0.9 [aA] | 78.6 ± 0.9 [aAB] | 77.5 ± 0.4 [aB] | 77.7 ± 0.3 [aB] |
| | 19 | 74.0 ± 0.4 [bC] | 78.6 ± 0.6 [aA] | 76.0 ± 0.6 [bB] | 75.3 ± 0.8 [bBC] | 74.3 ± 0.8 [bC] |
| | 21 | 73.3 ± 0.2 [bcB] | 76.2 ± 0.7 [bA] | 74.7 ± 0.7 [bAB] | 72.7 ± 0.1 [cC] | 73.1 ± 0.2 [bBC] |
| | 23 | 71.9 ± 0.4 [cBC] | 75.1 ± 0.7 [bA] | 72.5 ± 0.7 [cB] | 70.9 ± 0.4 [dC] | 69.7 ± 0.4 [cC] |
| | 25 | 69.3 ± 0.9 [dBC] | 72.6 ± 0.5 [cA] | 70.2 ± 0.5 [dB] | 68.6 ± 0.9 [eC] | 68.6 ± 0.6 [cC] |
| 2019 | 17 | 78.8 ± 0.2 [aB] | 81.0 ± 0.3 [aA] | 78.9 ± 0.7 [aB] | 78.7 ± 0.0 [aB] | 78.3 ± 0.0 [aB] |
| | 19 | 75.2 ± 0.2 [bBC] | 78.4 ± 0.9 [bA] | 76.5 ± 0.2 [bB] | 74.5 ± 0.1 [bC] | 75.1 ± 0.7 [bBC] |
| | 21 | 73.5 ± 1.3 [cB] | 75.2 ± 0.1 [bA] | 74.8 ± 0.6 [cAB] | 71.9 ± 0.6 [cC] | 71.8 ± 0.4 [cC] |
| | 23 | 70.9 ± 0.0 [dB] | 72.8 ± 0.3 [cA] | 71.5 ± 0.3 [dAB] | 69.7 ± 0.7 [dBC] | 68.4 ± 0.2 [dC] |
| | 25 | 69.3 ± 0.8 [eB] | 71.2 ± 0.7 [dA] | 69.6 ± 0.8 [eB] | 64.2 ± 0.9 [eD] | 67.0 ± 0.7 [eC] |

### 3.2. Kernel Sugars and Phytoglycogen Change during Maturation

According to consumer preferences, flavor is the most important indicator of sweet corn eating quality [7–9]. Sensory evaluations show a high correlation between the sweetness of sweet corn and the sugar content in the endosperm [10–12,32]. When assessing sweetness, it is important to consider all sugars because they may have different flavor intensities [33]. Kernel sucrose content in evaluated sweet corn hybrids prevailed over other sugars present, fructose, and glucose (Figure 1).

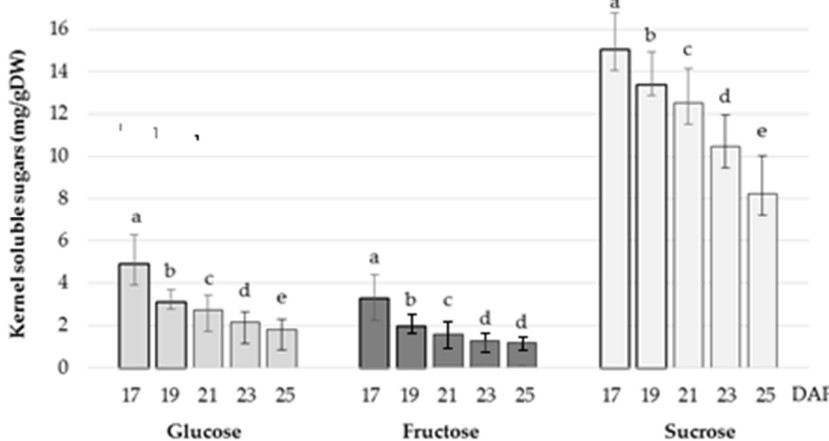

**Figure 1.** Kernel glucose, fructose and sucrose content changes during *su1* sweet corn maturation. Histograms show mean values ± standard deviation across five hybrids and two years. The different letters indicate significant differences among the harvest dates in each soluble sugar ($p <$ 0.05).

The proportion of sucrose in total soluble sugars across all hybrids ranged from 66.4% to 77.1%, which is less than 85% stated by Becerra-Sanchez and Taylor [33]. Generally, the content of each sugar as well as total sugars content decreased during maturation. However, significant genotype × harvest date × year interaction (Table S1) indicates that hybrids differed in the dynamics of sugars change in both years. Maximum total sugar content was obtained at 17 DAP, and for all hybrids it was significantly higher in 2019 (Table 3). The obtained results are in accordance with the research of Nemeskéri et al. [34], who found an increase of total sugars content in sweet corn under water stress. Hybrid OS 255su had the highest total sugars content in both years (21.3 and 28.7 mg/g DW). From first to last harvest date, total sugars reduction ranges from 27% (OS 230su) to 67% (OS 255su). Hybrid OS 230su had much higher sugars reduction in 2018 compared to 2019, while hybrid OS 255 showed the opposite reaction. Considering that commercial *su1* hybrids are characterized by total sugar content above 15 mg/g DW [6], only hybrid OS 230su in 2019 retained the required sweetness up to 23 DAP.

**Table 3.** Kernel total sugars content (mg/g DW) in five *su1* sweet corn hybrids at five harvest dates (17–25 DAP) in two growing seasons (Osijek, 2018 and 2019). The different lowercases show significant differences among the harvest dates within hybrids ($p < 0.05$). The different capital letters show significant differences among the hybrids ($p < 0.05$).

| Year | DAP | OS 254su | OS 255su | OS 256su | OS 247su | OS 230su |
|------|-----|----------|----------|----------|----------|----------|
|      | 17 | 18.9 ± 1.3 [aB] | 21.3 ± 2.5 [aA] | 16.5 ± 1.9 [aC] | 18.2 ± 1.1 [aB] | 20.0 ± 1.4 [aAB] |
|      | 19 | 18.7 ± 0.2 [aA] | 15.9 ± 0.4 [bB] | 14.3 ± 1.8 [bB] | 15.1 ± 0.7 [bB] | 18.6 ± 0.0 [aA] |
| 2018 | 21 | 15.3 ± 1.7 [bBC] | 15.6 ± 1.0 [bB] | 13.6 ± 0.4 [bC] | 12.2 ± 0.1 [cC] | 18.4 ± 0.2 [aA] |
|      | 23 | 13.1 ± 0.0 [cB] | 12.1 ± 0.6 [cB] | 13.5 ± 2.4 [bB] | 10.2 ± 0.3 [dC] | 16.4 ± 0.1 [bA] |
|      | 25 | 11.2 ± 0.3 [dB] | 11.8 ± 0.0 [cB] | 11.1 ± 0.9 [cB] | 7.5 ± 3.4 [eC] | 14.5 ± 0.7 [cA] |
|      | 17 | 24.5 ± 0.6 [aB] | 28.7 ± 1.7 [A] | 25.3 ± 0.0 [B] | 20.4 ± 0.3 [C] | 18.8 ± 1.7 [C] |
|      | 19 | 20.4 ± 0.1 [bA] | 19.5 ± 1.2 [A] | 19.9 ± 0.1 [A] | 15.3 ± 3.2 [B] | 15.4 ± 1.8 [B] |
| 2019 | 21 | 18.9 ± 2.4 [bA] | 17.9 ± 1.2 [AB] | 16.8 ± 0.4 [B] | 13.9 ± 2.0 [C] | 12.0 ± 0.9 [D] |
|      | 23 | 14.6 ± 1.1 [cA] | 13.0 ± 2.5 [A] | 13.1 ± 2.3 [A] | 12.8 ± 0.1 [B] | 11.1 ± 0.7 [B] |
|      | 25 | 11.5 ± 0.8 [dA] | 9.4 ± 0.4 [B] | 10.8 ± 0.3 [A] | 9.6 ± 0.8 [B] | 7.6 ± 0.6 [C] |

One of the *su1* sweet corn characteristics is a large amount of phytoglycogen that gives the kernel creamy texture [6]. Harvest date had a significant effect on phytoglyco-

gen while the effect of genotype and year were not significant (Table S1). As expected, due to the conversion of sugar to phytoglycogen, its content increased during maturation and reached a maximum concentration of 25 DAP (Table 4). Hybrid OS 247su had the largest change in phytoglycogen during maturation (from 10.1 to 25.0 mg/g DW). Creech et al. [13] found a similar accumulation of phytoglycogen in *su1* genotypes from 16 DAP to 22 DAP (from 14.3 to 28.5 mg/g DW) while further maturation to the 28 DAP did not significantly change its content.

**Table 4.** Kernel phytoglycogen content (%) in five *su1* sweet corn hybrids at five harvest dates (17–25 DAP) in two growing seasons (Osijek, 2018 and 2019). The different lowercases show significant differences among the harvest dates within hybrids ($p < 0.05$).

| Year | DAP | OS 254su | OS 255su | OS 256su | OS 247su | OS 230su |
|------|-----|----------|----------|----------|----------|----------|
|      | 17  | 16.4 ± 0.6 [a] | 15.5 ± 1.0 [a] | 17.4 ± 0.8 [a] | 10.1 ± 1.2 [a] | 16.0 ± 0.6 [a] |
|      | 19  | 18.5 ± 0.1 [b] | 19.7 ± 0.2 [b] | 20.3 ± 0.7 [b] | 20.0 ± 0.3 [b] | 18.2 ± 0.5 [b] |
| 2018 | 21  | 19.9 ± 0.7 [b] | 19.8 ± 0.4 [b] | 20.6 ± 0.2 [b] | 21.1 ± 0.1 [b] | 18.7 ± 0.1 [b] |
|      | 23  | 22.8 ± 0.1 [c] | 23.2 ± 0.2 [cb] | 22.6 ± 1.0 [c] | 23.9 ± 0.4 [c] | 21.5 ± 0.1 [c] |
|      | 25  | 23.5 ± 0.1 [c] | 23.3 ± 0.1 [c] | 23.6 ± 0.4 [c] | 25.0 ± 1.4 [c] | 22.2 ± 0.3 [c] |
|      | 17  | 14.2 ± 0.3 [a] | 12.5 ± 0.7 [a] | 13.9 ± 0.0 [a] | 15.8 ± 0.1 [a] | 16.5 ± 0.7 [a] |
|      | 19  | 17.8 ± 0.0 [b] | 18.2 ± 0.5 [b] | 18.0 ± 0.1 [b] | 19.9 ± 1.3 [b] | 19.8 ± 0.8 [b] |
| 2019 | 21  | 18.5 ± 1.0 [b] | 18.9 ± 0.5 [b] | 19.3 ± 0.2 [b] | 20.5 ± 0.8 [b] | 21.2 ± 0.4 [b] |
|      | 23  | 22.3 ± 0.5 [c] | 22.8 ± 1.0 [c] | 22.8 ± 0.9 [c] | 23.0 ± 0.1 [c] | 23.6 ± 0.3 [c] |
|      | 25  | 23.4 ± 0.4 [c] | 24.3 ± 0.2 [c] | 23.7 ±0.1 [c] | 24.2 ± 0.3 [c] | 25.0 ± 0.3 [d] |

*3.3. Kernel total phenols and antioxidant activity change during maturation*

Kernel total phenols content is an important factor in the health and functional value of sweet corn [20,21,35]. Tested hybrids differed significantly in total phenols (204.4–357.1 mg GAE/100 g DW) (Table 5). Reduction in total phenols content during maturation ranged among hybrids from 17.4% (OS 254su) to 29.3% (OS 256su). The results are not consistent with studies that have shown a significant increase in total phenols content during sweet corn maturation [36–38]. The mismatch of results may be due to different genotypes (particularly different mutant type) and different periods of sweet corn maturity examined in those studies. Maximum total phenols content as well as its decrease was higher in 2019, which could be due to extreme drought conditions during maturation. However, the change of total phenols during maturation for hybrids OS 256su and Os 247su in 2018 was not significant.

**Table 5.** Kernel total phenols content (mg GAE/100 g DW) in five *su1* sweet corn hybrids at five harvest dates (17–25 DAP) in two growing seasons (Osijek, 2018 and 2019). The different lowercases show significant differences among the harvest dates within hybrids ($p < 0.05$). The different capital letters show significant differences among the hybrids ($p < 0.05$).

| Year | DAP | OS 254su | OS 255su | OS 256su | OS 247su | OS 230su |
|------|-----|----------|----------|----------|----------|----------|
|      | 17  | 254 ± 37 [aB] | 204 ± 37 [aC] | 238 ± 32 [aB] | 252 ± 2 [aB] | 291 ± 17 [aA] |
|      | 19  | 255 ± 2 [aA] | 167 ± 1 [bC] | 241 ± 8 [aB] | 241 ± 25 [aB] | 286 ± 24 [aA] |
| 2018 | 21  | 219 ± 31 [bB] | 181 ± 12 [abC] | 222 ± 3 [aB] | 246 ± 13 [aB] | 288 ± 25 [aA] |
|      | 23  | 220 ± 31 [bB] | 180 ± 33 [abB] | 238 ± 1 [aA] | 235 ± 32 [aA] | 257 ± 33 [bA] |
|      | 25  | 244 ± 6 [abA] | 159 ± 12 [bC] | 220 ± 6 [aB] | 245 ± 57 [aA] | 258 ± 16 [bA] |
|      | 17  | 278 ± 0 [aB] | 223 ± 19 [aC] | 331 ± 27 [aA] | 283 ± 13 [aB] | 357 ± 9 [aA] |
|      | 19  | 250 ± 13 [abB] | 186 ± 28 [bC] | 300 ± 9 [bA] | 249 ± 0 [bB] | 285 ± 0 [bA] |
| 2019 | 21  | 252 ± 22 [abA] | 184 ± 3 [bB] | 277 ± 4 [bA] | 243 ± 7 [bA] | 272 ± 18 [bA] |
|      | 23  | 245 ± 23 [bB] | 173 ± 14 [bC] | 234 ± 14 [cB] | 230 ± 0 [Bb] | 282 ± 1 [bA] |
|      | 25  | 229 ± 4 [bB] | 178 ± 2 [bC] | 246 ± 0 [cB] | 229 ± 0 [bB] | 286 ± 9 [bA] |

To date, various methods have been applied to measuring the antioxidant activity of sweet corn kernels. We used DPPH assay, which is based on hydrogen atom transfer reaction, and it has been widely used for assessment of radical scavenging activity [39,40]. Previous studies showed the inconsistency in antioxidant activity obtained for sweet corn by different methods [35,41,42]. The DPPH scavenging activity determined in sweet corn vary considerably (29.1–83.6% inhibition) among studies [22,35,37,43,44]. According to Khampas et al. [41], those differences could be due to a genetic background, stage of maturity or wide range of environmental factors which affect antioxidant activity. In our study, DPPH scavenging activity of *su1* sweet corn genotypes ranged from 52.5% to 87.1% inhibition (Table 6) and was significantly affected by harvest date. In generally, hybrids increased antioxidant activity during maturation and larger increase was obtained in 2019. Since hybrids showed a completely opposite reaction for total phenols content, it can be assumed that other phytochemicals in addition to phenols also played an important role in antioxidant activity, especially in a stressful environment. Hybrid OS 255su had the lowest variation in antioxidant activity across harvest dates and years.

**Table 6.** DPPH radical scavenging activity (%) in five *su1* sweet corn hybrids at five harvest dates (17–25 DAP) in two growing seasons (Osijek, 2018 and 2019). The different lowercases show significant differences among the harvest dates within hybrids ($p < 0.05$). The different capital letters show significant differences among the hybrids ($p < 0.05$).

| Year | DAP | OS 254su | OS 255su | OS 256su | OS 247su | OS 230su |
|---|---|---|---|---|---|---|
| 2018 | 17 | 64.8 ± 21.9 aB | 73.8 ± 13.2 bA | 74.9 ± 11.6 aA | 68.1 ± 26.0 aB | 52.5 ± 26.6 aC |
| | 19 | 64.1 ± 19.7 aB | 67.5 ± 10.1 aAB | 70.1 ± 5.0 aA | 69.2 ± 15.0 aAB | 66.2 ± 20.7 bAB |
| | 21 | 78.2 ± 13.7 bB | 86.5 ± 14.2 cA | 74.8 ± 2.0 aB | 66.4 ± 14.5 bC | 69.3 ± 4.8 bC |
| | 23 | 70.2 ± 14.6 abB | 76.3 ± 19.2 bA | 80.0 ± 6.5 bA | 74.7 ± 25.3 bAB | 67.8 ± 4.3 bB |
| | 25 | 79.7 ± 23.4 bA | 84.2 ± 6.1 cA | 80.7 ± 22.6 bA | 77.9 ± 26.6 bB | 68.2 ± 21.3 bC |
| 2019 | 17 | 55.9 ± 6.4 aC | 72.1 ± 15.6 aA | 62.8 ± 11.6 aB | 68.8 ± 12.4 aAB | 58.3 ± 5.3 aBC |
| | 19 | 71.6 ± 6.3 bB | 80.5 ± 1.9 bA | 67.7 ± 13.7 aB | 78.0 ± 4.1 bA | 69.8 ± 9.0 bB |
| | 21 | 78.5 ± 2.7 bcAB | 83.1 ± 0.3 bA | 75.5 ± 2.7 bB | 78.7 ± 0.4 bAB | 74.4 ± 6.6 bcB |
| | 23 | 82.9 ± 1.5 cA | 82.8 ± 1.3 bA | 74.2 ± 7.5 bB | 81.3 ± 1.4 bA | 83.9 ± 3.5 cA |
| | 25 | 83.6 ± 1.7 cAB | 81.3 ± 6.5 bAB | 78.2 ± 10.9 bB | 80.9 ± 0.1 bB | 87.1 ± 1.6 cA |

## 4. Conclusions

Our study has confirmed that the kernel quality of *su1* sweet corn hybrids changes significantly during maturation. Rapid decrease in kernel moisture and sugars content resulted in a narrowed optimal harvest window. Depending on the hybrid, the duration of the optimal harvest period was from two to four successive harvest dates, i.e., it lasted 2 to 6 vegetation days. Based on the obtained results, hybrids with higher sugar content and also slower kernel moisture and sugars loss can be recommended to producers.

Within a short optimal harvest period, a significant change in kernel total phenols and antioxidant activity was determined. Although the differences among hybrids for total phenols and antioxidant activity have been identified, it is not possible to highlight hybrids with potentially higher health benefits due to high hybrid variation across harvest dates and years. Better assessment could be given by examining a larger number of genotypes. However, high total phenols content and high atioxdant activity in *su1* sweet corn shoud have an impact on both producers and consumers by encouraging them to shift their preferences from sensory enjoyment towards health benefits.

**Supplementary Materials:** The following are available online at www.mdpi.com/article/10.3390/agronomy12051224/s1, Table S1: Mean squares from the ANOVA for the examined components of kernel quality in su1 sweet corn genotypes.

**Author Contributions:** Conceptualization, T.L., Z.S., D.Š; methodology D.H., D.Š.; software, Z.S.; formal analysis, T.L., D.H.; investigation, T.L., S.Š.M., D.H.; resources, A.J. and D.Š.; data curation, T.L., D.H.; writing—original draft preparation, T.L; writing—review and editing, S.Š.M., D.H., Z.S., D.Š.; funding acquisition, D.Š. All authors have read and agreed to the published version of the manuscript.

**Funding:** Centre of Excellence for Biodiversity and Molecular Plant Breeding, Zagreb, Croatia: KK.01.1.1.01.0005.

**Institutional Review Board Statement:**　Not applicable.

**Informed Consent Statement:**　Not applicable.

**Data Availability Statement:**　Publicly available weather data were reported in this study.

Data can be found at: https://klima.hr/razno/publikacije/agroklimatski_atlas_RH_1981_2020.pdf (accessed on 2 March 2022).

**Conflicts of Interest:** The authors declare no conflict of interest.

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
