# Peer review of "Effect of Harvest Date on Kernel Quality and Antioxidant Activity in su1 Sweet Corn Genotypes"

_agronomy, doi:10.3390/agronomy12051224_

Round 1
Reviewer 1 Report
A very interesting paper dealing with the quality of sweet maize that is one of the most important and most valuable vegetable in the region.
In general the only confusing issue is that the kernel moisture is a parameter appears several times, but it is a state not a quality parameter as it is changing any time related to the environmental parameters and so on. If you mean ‘kernel moisture at harvest’ with it, than use that term anywhere!
line 16: su1 should be in italic style
line 18: five harvest by year
line 31: what do you mean with ‘field corn’?
line 33: su1, sh2 and se1 should be in italic style
line 40: kernel moisture at harvest – in any cases in the following parts as well
line 43: 76% and 70% instead of 76 % and 70 %
line 55: not ‘several times lower’ but just ‘significantly lower’
line 87: 10.8 oC instead of 10.8oC
line 87: precipitation instead of rainfall
line 111: 4 oC
line 117-118: to avoid splitting the number from the unit, you can use non-breaking space (ctrl + shift + space)
line 124: ml is the official not mL, also further in the text
line 132: µl instead of μL
line 139: the dry weight is not the first place abbreviated as DW, define it there!
line 152: what was the concentration of methanol?
line 158: correct the formulas!
line 159: RT means room temperature?
line 172: does it mean the blank was not incubated for 30 min in dark? It is not clear!
line 175: applied
line 176: genotype
line 212: the standard deviation in + and – is symmetric, so there should be a mistake as for example in fructose 23 and 25 it is visibly not true. Correct it!
line 224: there is no capital letter in the table. If it is just missing, insert them!
line 251: 17.4%
line 316: scientific name in italic!
line 326: su, se and sh2 in italic
line 372: scientific name in italic!
line 383: scientific name in italic!
line 386: scientific name in italic!
line 395: scientific name in italic!
line 398: scientific name in italic!
Author Response
Thank you for your review. These are our responses to your comments.
In general the only confusing issue is that the kernel moisture is a parameter appears several times, but it is a state not a quality parameter as it is changing any time related to the environmental parameters and so on. If you mean ‘kernel moisture at harvest’ with it, than use that term anywhere!
RESPONSE: We changed it where appropriate.
line 16: su1 should be in italic style
RESPONSE: changed
line 18: five harvest by year
RESPONSE: changed
line 31: what do you mean with ‘field corn’?
RESPONSE: added “(flint/dent)”
line 33: su1, sh2 and se1 should be in italic style
RESPONSE: changed
line 40: kernel moisture at harvest – in any cases in the following parts as well
RESPONSE: changed where appropriate
line 43: 76% and 70% instead of 76 % and 70 %
RESPONSE: changed
line 55: not ‘several times lower’ but just ‘significantly lower’
RESPONSE: changed
line 87: 10.8 oC instead of 10.8oC
RESPONSE: changed
line 87: precipitation instead of rainfall
RESPONSE: changed
line 111: 4 oC
RESPONSE: changed
line 117-118: to avoid splitting the number from the unit, you can use non-breaking space (ctrl + shift + space)
RESPONSE: used
line 124: ml is the official not mL, also further in the text
RESPONSE: changed throughout
line 132: µl instead of μL
RESPONSE: changed
line 139: the dry weight is not the first place abbreviated as DW, define it there!
RESPONSE: changed
line 152: what was the concentration of methanol?
RESPONSE: concentration added
line 158: correct the formulas!
RESPONSE: corrected
line 159: RT means room temperature?
RESPONSE: changed
line 172: does it mean the blank was not incubated for 30 min in dark? It is not clear!
RESPONSE: A new sentence added.
line 175: applied
RESPONSE: changed
line 176: genotype
RESPONSE: changed
line 212: the standard deviation in + and – is symmetric, so there should be a mistake as for example in fructose 23 and 25 it is visibly not true. Correct it!
RESPONSE: corrected
line 224: there is no capital letter in the table. If it is just missing, insert them!
RESPONSE: corrected. One sentence in caption is deleted.
line 251: 17.4%
RESPONSE: changed
line 316: scientific name in italic!
RESPONSE: changed
line 326: su, se and sh2 in italic
RESPONSE: changed
line 372: scientific name in italic!
RESPONSE: changed
line 383: scientific name in italic!
RESPONSE: changed
line 386: scientific name in italic!
RESPONSE: changed
line 395: scientific name in italic!
RESPONSE: changed
line 398: scientific name in italic!
RESPONSE: changed
Reviewer 2 Report
The paper examines effects of harvest dates on kernel quality and antioxidant activity of different sweet corn genotypes. From this aspects, the subject of the paper is relevant, informative, interesting and within scope of the journal.
Introduction section of the manuscript clearly describes importance of sweet corn for vegetable production and the effects of the maturation stage on sweet corn quality. The aim of the study was also clearly described at the end of the introduction section.
The materials and methods used in the study were clearly described in the manuscript. Standard methods were given by references.
Results were presented in a logical order. The findings of the study were satisfactorily discussed in the related section.
Writing style of the manuscript is academic and clear. In my opinion, originality and common influence of the article are quite high, so the manuscript can be accepted for publication after minor revision.
My further comments are included below.
Line 58. Replace extend with extended.
Line 91-92. The authors indicated that the experiment was conducted according to RCBD but Table S1, 2, 3, 4, 5 and 6 shows that the data were analysed according to split plot design. Additionally, please clarify which treatments are main or sub-plots.
Table 1. Total precipitation and averaged temperature data should be added.
Line 187 – 188. Genotypes reached …… 17 – 19 DAP. This suggestion is questionable because all cultivars had moisture contents above the threshold (70 – 76%) at 17 DAP. Additionally, most of the cultivars had optimum moisture contents between 19 – 23 DAP during the study.
Line 248 – 250. Tested hybrids ……….obtained at 17 DAP (Table 5). This comment is wrong. Exceptionally, OS 254su and OS 256su hybrids had higher phenols contents at 19 DAP than at 17 DAP.
Author Response
Thank you for your review. These are our responses to your comments.
Line 58. Replace extend with extended.
RESPONSE: replaced
Line 91-92. The authors indicated that the experiment was conducted according to RCBD but Table S1, 2, 3, 4, 5 and 6 shows that the data were analysed according to split plot design. Additionally, please clarify which treatments are main or sub-plots.
RESPONSE: The subsection “2.7 Statistical analysis” is rewritten.
Table 1. Total precipitation and averaged temperature data should be added.
RESPONSE: Total precipitation and averaged temperature are added.
Line 187 – 188. Genotypes reached …… 17 – 19 DAP. This suggestion is questionable because all cultivars had moisture contents above the threshold (70 – 76%) at 17 DAP. Additionally, most of the cultivars had optimum moisture contents between 19 – 23 DAP during the study.
RESPONSE: The sentence is deleted.
Line 248 – 250. Tested hybrids ……….obtained at 17 DAP (Table 5). This comment is wrong. Exceptionally, OS 254su and OS 256su hybrids had higher phenols contents at 19 DAP than at 17 DAP.
RESPONSE: This part of the sentence is deleted.